# Synaptic Plasticity-Enhancing and Cognitive-Improving Effects of Standardized Ethanol Extract of *Perilla frutescens* var. *acuta* in a Scopolamine-Induced Mouse Model

**DOI:** 10.3390/ijms26209925

**Published:** 2025-10-12

**Authors:** Jihye Lee, Eunhong Lee, Hyunji Kwon, Somin Moon, Ho Jung Bae, Joon-Ho Hwang, Gun Hee Cho, Haram Kong, Mi-Houn Park, Sung-Kyu Kim, Dong Hyun Kim, Ji Wook Jung

**Affiliations:** 1Industry-Academic Cooperation Foundation, Daegu Haany University, 1 Hanuidae-ro, Gyeongsan 38610, Gyeongsangbuk-do, Republic of Korea; jhlee86@dhu.ac.kr; 2School of Chemical Engineering, Yeungnam University, 280 Daehak-Ro, Gyeonngsan 38541, Gyeongsangbuk-do, Republic of Korea; lablabuuu@naver.com; 3Department of Advanced Translational Medicine, School of Medicine, Konkuk University, 120 Neungdong-ro, Gwangjin-gu, Seoul 05029, Republic of Korea; hyunje2701@naver.com (H.K.); ans5346@naver.com (S.M.); 4Department of Bio Health Science, College of Natural Science, Changwon National University, Changwon 51140, Gyeongsangnam-do, Republic of Korea; baehj321@changwon.ac.kr; 5Borambio Co., Ltd., 119, Dandae-ro, Dongnam-gu, Cheonan 31116, Chungcheongnam-do, Republic of Korea; jh_hwang@borambio.com (J.-H.H.); joo2413@borambio.com (G.H.C.); hrkong@borambio.com (H.K.); skkim@borambio.com (S.-K.K.); 6Department of Pharmacology, School of Medicine, Konkuk University, 268 Chungwon-daero, Chungju-si 27478, Chungcheongbuk-do, Republic of Korea; 7Research Institute of Medical Science, Konkuk University School of Medicine, 268 Chungwon-daero, Chungju-si 27478, Chungcheongbuk-do, Republic of Korea; 8Department of Cosmetics, College of K-Bio Health, Daegu Haany University, 285-10, Eobongji-gil, Gyeongsan 38578, Gyeongsangbuk-do, Republic of Korea

**Keywords:** learning and memory, cognitive impairments, synaptic plasticity, muscarinic acetylcholine receptor, *Perilla frutescens* var. *acuta*

## Abstract

In our previous study, we demonstrated that a standardized ethanol extract of *Perilla frutescens* var. *acuta* (PE) alleviates memory deficits in an Alzheimer’s disease mouse model by inhibiting amyloid β (Aβ) aggregation and promoting its disaggregation. However, the extent to which PE exerts additional cognitive benefits independent of Aβ pathology remained unclear. Here, we aimed to evaluate the effects of PE on synaptic plasticity and learning and memory functions. Male ICR mice were used, and cognitive impairment was induced by scopolamine administration. PE was orally administered at doses determined from previous studies, and cognitive performance was assessed using the passive avoidance, Y-maze, and Morris water maze tests. In parallel, hippocampal slices were employed to examine the effects of PE on synaptic plasticity. PE (100 and 300 μg/mL) significantly enhanced long-term potentiation (LTP) in a concentration-dependent manner without altering basal synaptic transmission. This facilitation of LTP was blocked by scopolamine (1 μM), a muscarinic acetylcholine receptor (mAChR) antagonist, and IEM-1460 (50 μM), a calcium-permeable α-amino-3-hydroxy-5-methyl-4-isoxazolepropionic acid receptor (CP-AMPAR) inhibitor, indicating the involvement of mAChR and CP-AMPAR pathways. In vivo, PE (100, 250, and 500 mg/kg) treatment improved memory performance across all behavioral tasks and upregulated hippocampal synaptic proteins including GluN2B, PSD-95, and CaMKII. Collectively, these results demonstrate that PE ameliorates scopolamine (1 mg/kg)-induced cognitive impairment by enhancing synaptic plasticity, likely through modulation of mAChR, CP-AMPAR, and NMDA receptor signaling. These findings highlight the therapeutic potential of PE for memory deficits associated with cholinergic dysfunction.

## 1. Introduction

In modern society, cognitive function is regarded as a fundamental element that determines not only the efficiency of daily life activities but also an individual’s overall quality of life. Cognitive function encompasses various mental abilities, including attention, memory, executive function, language comprehension, and judgment, all of which enable individuals to acquire new information, solve problems, and make appropriate decisions [1]. However, cognitive function can deteriorate either gradually or rapidly due to factors such as aging, stress, neurological disorders, or adverse drug effects [2,3]. Such decline can significantly impair an individual’s autonomy and social functioning.

Cognitive impairment can arise from various reasons, including neurotransmitter imbalances [4], decreased receptor density [5], inflammation [6], and oxidative stress [7], leading to synaptic dysfunction. Nevertheless, a commonality among these factors is the impairment of synaptic plasticity, particularly in the hippocampus, a crucial brain region known for its cellular mechanisms underlying learning and memory, where long-term potentiation (LTP) plays a pivotal role [8,9]. Aberrations in LTP are associated with cognitive decline, highlighting the importance of identifying substances that can modulate LTP in the hippocampus as a potentially expedited approach to developing drugs for improving cognition.

Recent research has indicated that standardized ethanol extract of *Perilla frutescens* var. *acuta* (L.) Britt. leaves (PE), a member of the Lamiaceae family, can alleviate symptoms in Alzheimer’s disease mouse models through its ability to inhibit amyloid β (Aβ) aggregation [10]. Moreover, recent studies have reported that *P. frutescens* extracts exert antioxidant, anti-inflammatory, and neuroprotective effects that may contribute to improved cognitive function. For example, *P. frutescens* leaf extract was shown to ameliorate cognitive impairment and protect against neuronal damage in Alzheimer’s disease models [11], while dietary supplementation with *P. frutescens* attenuated neuroinflammation and improved memory-related behavior [12]. PE contains a variety of polyphenols, including flavonoids, which have been demonstrated through numerous studies to influence synaptic function [13,14,15]. Therefore, in this study, we evaluated the effects of PE on hippocampal LTP and investigated its potential for enhancing memory in a well-established cognitive impairment animal model induced by scopolamine. The use of ICR mice and the scopolamine-induced cognitive impairment model was appropriate to address our objectives. ICR mice are commonly used in behavioral studies for their reliability, and the scopolamine model mimics cholinergic dysfunction, a key feature of human memory impairment. Therefore, this approach provides translational relevance to human biology.

This study aimed to investigate whether standardized ethanol extract of *Perilla frutescens* var. *acuta* (PE) enhances hippocampal synaptic plasticity and ameliorates scopolamine-induced memory impairments and to explore the potential mechanisms involved.

## 2. Results

### 2.1. Effect of PE on Hippocampal LTP

To test the effect of PE on hippocampal LTP, acute hippocampal slices were incubated with PE (30, 100, or 300 μg/mL) for 2 h. Then, LTP was induced by HFS after 20 min period of stable baseline. In the control group, HFS induced approximately 123 ± 2 of LTP (Figure 1A,B). In the group treated with 100 or 300 μg/mL of PE, approximately 140 ± 2 or 151 ± 8 of LTP was induced, respectively, which was statistically significantly different from the control group (*F*_3, 20_ = 11.03, *p* < 0.05, n = 6/group, Figure 1B). This suggests that PE can increase hippocampal LTP. To determine whether PE had an impact on basal synaptic transmission, we measured PPR (Figure 1C) and the input-output curve (Figure 1D) before and after HFS. However, PE had no significant effect on these basal synaptic transmissions.

### 2.2. Involvement of mAChR and Calcium-Permeable AMPA Receptor on the Effect of PE on Hippocampal LTP

Various substances with cognitive enhancement effects have mechanisms that increase acetylcholine levels [16,17]. Although scopolamine, a muscarinic acetylcholine receptor (mAChR) antagonist, inhibited the LTP-increasing effect of PE (two-way ANOVA, PE, *F*_1, 30_ = 9.725, *p* < 0.05; scopolamine, *F*_1, 30_ = 0.3822, *p* > 0.05; interaction, *F*_1, 30_ = 4.775, *p* < 0.05, n = 7–11/group, Figure 2A,B), PE induced the LTP increase, significantly. The LTP increase induced by mAChR stimulation is associated with an increase in calcium-permeable AMPA receptors at the synapse [18]. To investigate whether PE induces an LTP increase through calcium-permeable AMPA receptors (CP-AMPAR), we applied IEM-1460. When IEM-1460 was administered during the baseline period, it was observed that the LTP increase induced by PE disappeared (two-way ANOVA, PE, *F*_1, 24_ = 5.636, *p* < 0.05; scopolamine, *F*_1, 24_ = 6.265, *p* < 0.05; interaction, *F*_1,24_ = 17.27, *p* < 0.05, n = 7/group, Figure 2C,D).

### 2.3. Effect of PE on Scopolamine-Induced Cognitive Impairment in Passive Avoidance Test

Next, to determine whether the LTP-enhancing effect of PE could also be observed in vivo, we examined the effect of PE in a scopolamine-induced cognitive impairment model. In an acquisition trial, there were no significant differences in latency time among all groups (*F*_5, 30_ = 0.9171, *p* > 0.05, n = 6/group, Figure 3). Scopolamine administration in the passive avoidance test resulted in a significant decrease in latency time of the retention trial (Figure 3, *p* < 0.05). This indicates that scopolamine induced cognitive impairment. However, in the group treated with PE (250 or 500 mg/kg, p.o.), the cognitive impairment induced by scopolamine was not observed (*F*_5, 30_ = 7.568, *p* < 0.05, n = 6/group, Figure 3). Donepezil (DNPZ), a positive drug, also blocked scopolamine-induced cognitive impairment in the passive avoidance test (Figure 3, *p* < 0.05). These results indicated that PE improves scopolamine-induced cognitive impairment in the passive avoidance test without affecting normal movement.

### 2.4. Effect of PE on Scopolamine-Induced Memory Impairment in Y-Maze Test

Next, to confirm whether PE improves scopolamine-induced cognitive impairment in another memory assessment experiment, we conducted the Y-maze test. The administration of scopolamine significantly decreased spontaneous alternation (Figure 4A, *p* < 0.05), indicating that scopolamine induced a decline in working memory. However, in the group treated with PE (100, 250, or 500 mg/lg, p.o.), the cognitive impairment induced by scopolamine was not observed (*F*_5, 44_ = 6.343, *p* < 0.05, n = 8–9/group, Figure 4A). Donepezil, a positive drug, also blocked scopolamine-induced memory impairment in the Y-maze test (Figure 4A, *p* < 0.05). PE and donepezil did not have an impact on total arm entry (*F*_5, 44_ = 3.030, *p* > 0.05, n = 8–9/group, Figure 4B). These results suggest that PE improves scopolamine-induced cognitive impairment in the Y-maze test without affecting movement.

### 2.5. Effect of PE on Scopolamine-Induced Cognitive Impairment in Morris Water Maze Test

To assess the effects of PE on spatial memory, the Morris water maze test was conducted. During the four-day training period, the group treated with scopolamine showed a significant increase in the time taken to find the platform (Figure 5A, *p* < 0.05). The administration of PE demonstrated an inhibitory effect on this increase in latency time (two-way ANOVA, day, *F*_3, 144_ = 38.60, *p* < 0.05; group, *F*_3, 144_ = 14.98, *p* < 0.05; interaction, *F*_15, 144_ = 0.9315, *p* > 0.05, n = 7/group, Figure 5A). In the probe test, the scopolamine-treated group showed a significant decrease in the time spent swimming in the quadrant where the platform had been located (Figure 5B, *p* < 0.05). PE administration was able to suppress this decrease (*F*_5, 36_ = 5.088, *p* < 0.05, n = 7/group, Figure 5B). These results suggest that PE improves scopolamine-induced cognitive impairment in the Morris water maze test.

### 2.6. Effect of PE on Glutamatergic Synapse

To assess the effects of PE on glutamatergic synapse, hippocampal level of GluN2B, a subunit of the NMDA receptor, PSD-95, a synaptic marker, and CaMKII were observed. In the scopolamine-treated group, there was a tendency for decreased expression of GluN2B (*p* > 0.05, Figure 6A,B) and PSD-95 (*p* > 0.05, Figure 6A,C); however, the differences were not statistically significant. In contrast, PE treatment significantly increased the expression of GluN2B (*F*_2, 9_ = 12.14, *p* < 0.05, n = 4/group, Figure 6A,B), PSD-95 (*F*_2, 9_ = 4.924, *p* < 0.05, n = 4/group, Figure 6A,C) and CaMKII (*F*_2, 9_ = 14.54, *p* < 0.05, n = 4/group, Figure 6A,D) in the hippocampus compared to the scopolamine-only group. These results suggest that PE may promote the enhancement of glutamatergic synapses.

## 3. Discussion

In this study, we demonstrated that PE enhances synaptic plasticity through mAChR signaling and alleviates scopolamine-induced cognitive impairment. Furthermore, PE increased the levels of synaptic proteins in the hippocampus of experimental animals, indicating an enhancement in synaptic function. These findings suggest that PE improves scopolamine-induced cognitive impairments, and this effect is likely associated with its ability to enhance synaptic plasticity. Previous studies have reported that natural compounds, including extracts of Perilla frutescens, exhibit inhibitory effects on Aβ aggregation and reduce amyloid plaque formation in Alzheimer’s disease models [10,11]; our study reveals that PE also has direct effects on synaptic plasticity and memory beyond its anti-amyloid effects (Figure 7).

mAChR stimulation is known to play a crucial role in memory formation in the hippocampus, and the blockade of these receptors is associated with impaired memory formation [19,20,21]. One of the key pathological mechanisms of AD is the reduction in acetylcholine signaling, which is associated with widespread neuronal loss. In particular, the degeneration of cholinergic neurons in the basal forebrain leads to decreased acetylcholine levels in the cerebral cortex and hippocampus, contributing to deficits in learning and memory [22]. Consequently, strategies aimed at preserving or enhancing acetylcholine signaling are considered crucial for improving cognitive function. Previous studies have reported that acetylcholinesterase inhibitors, which increase acetylcholine levels, improve memory by overcoming the blockade of mAChR [23]. Additionally, scopolamine is known to induce memory impairment through its interference with cholinergic neurotransmission, as well as by causing brain inflammation and oxidative damage [24,25,26]. Our study demonstrates that PE ameliorates scopolamine-induced memory impairments, likely through mAChR stimulation and its related effects on synaptic plasticity.

CP-AMPARs are a subtype of AMPA receptors that play a crucial role in synaptic plasticity, particularly in the induction of LTP, by regulating intracellular calcium influx [27]. These receptors are composed solely of GluR1 subunits, and their synaptic expression is modulated by the phosphorylation of GluR1. Previous studies have reported that acetylcholine facilitates GluR1 phosphorylation via mAChRs, thereby promoting the synaptic incorporation of CP-AMPARs and contributing to LTP formation [28]. Therefore, the reduced acetylcholine signaling is expected to impair LTP formation [29]. Increasing acetylcholine levels would be the most straightforward approach to overcoming this deficit; however, in this study, PE did not exhibit any effect on acetylcholine levels. Nevertheless, the finding that PE enhances LTP through mAChR and CP-AMPAR suggests that certain components of PE may influence the mAChR-mediated trafficking pathway of CP-AMPARs to synapses.

Changes in synaptic function through neuronal excitability are mediated by the expression levels of synaptic scaffold proteins such as PSD-95 [30]. PSD-95 regulates the trafficking and localization of AMPA and NMDA receptors. Overexpression of PSD-95 increases the amplitude of AMPA receptor-mediated synaptic currents, thereby mimicking the effects of LTP [31]. GluN2B, a subunit of NMDAR, plays a critical role in the induction of LTP. Overexpression of GluN2B in transgenic mice has been reported to enhance memory. GluN2B-containing NMDARs are strongly associated with CaMKII, a key protein involved in the expression of LTP as well as in learning and memory processes [32]. When calcium ions enter the postsynaptic neuron via CP-AMPAR or NMDA receptors, they form a Ca^2+^/calmodulin complex [33]. This complex binds to CaMKII and induces its autophosphorylation. Activated CaMKII translocates into the nucleus and promotes the upregulation of synaptic and signaling-related genes. The resulting mRNAs are transported to the cytoplasm, leading to the expression of proteins involved in synaptic plasticity. In our study, PE treatment induced the increased expression of PSD-95 and CaMKII in the scopolamine-induced cognitive impairment model. These findings suggest that PE may enhance synaptic plasticity by modulating key molecules involved in excitatory synaptic transmission.

PE contains a substantial amount of rosmarinic acid, which is known for its antioxidative [34], anti-inflammatory [35], neuroprotective [36], and neurogenesis-promoting properties [13] in various cognitive impairment models. Our supplementary results further demonstrated that rosmarinic acid significantly enhanced hippocampal LTP (Appendix A), supporting its potential contribution to synaptic plasticity. However, the specific effects of rosmarinic acid on synaptic plasticity and CP-AMPAR have not been extensively studied, and we cannot be certain that it is solely responsible for PE’s memory-improving effects. Considering that PE contains multiple bioactive constituents, the overall effects are likely mediated by a combination of compounds acting synergistically. Nevertheless, based on our previous findings in Alzheimer’s disease and Aβ-induced cognitive impairment models, we hypothesize that rosmarinic acid plays a significant role. Further research focusing on the role of rosmarinic acid and its influence on AMPAR, as well as investigations of other constituents in PE, will provide stronger evidence to support this hypothesis.

PE significantly improved scopolamine-induced cognitive impairment across passive avoidance, Y-maze, and Morris water maze tests without affecting spontaneous locomotion. Although we did not perform a dedicated open field test, the absence of significant changes in total arm entries during the Y-maze test and swimming performance in the Morris water maze suggests that PE did not affect general locomotor activity. These findings suggest that PE may counteract cholinergic dysfunction-induced cognitive deficits, particularly in hippocampus-dependent memory tasks such as spatial learning. Since scopolamine impairs memory through muscarinic receptor blockade, the reversal effect of PE may involve modulation of cholinergic neurotransmission. Moreover, the absence of changes in total arm entries or locomotor activity supports its selective action on cognition rather than general arousal or motor function.

Our results demonstrated that PE increased the expression of PSD-95, CaMKII, and GluN2B, which are known to interact in a coordinated manner to enhance synaptic plasticity. PSD-95 serves as a scaffold protein that regulates the localization of both AMPARs and NMDARs at the postsynaptic density, thereby stabilizing synaptic transmission. The upregulation of GluN2B-containing NMDARs facilitates Ca^2+^ influx, which subsequently activates CaMKII, a key kinase that promotes LTP expression and memory formation. The observed concurrent increase in these markers suggests that PE may strengthen excitatory synaptic connections through multiple converging mechanisms. These findings are consistent with previous reports showing that natural polyphenols, including rosmarinic acid, exert neuroprotective and cognition-enhancing effects via modulation of synaptic proteins and cholinergic neurotransmission. Thus, our study adds new evidence that PE improves synaptic plasticity by simultaneously targeting receptor function, synaptic scaffolding, and downstream signaling pathways.

This study has certain limitations. First, although we demonstrated the involvement of synaptic proteins such as PSD-95, CaMKII, and GluN2B, we did not measure other important markers including synaptophysin and BDNF, which would provide further validation of synaptic plasticity enhancement. Second, we inferred the role of calcium signaling through CP-AMPAR inhibition experiments but did not directly quantify intracellular Ca^2+^ levels. Future studies are warranted to address these limitations by incorporating synaptophysin/BDNF expression and calcium imaging to further strengthen the mechanistic understanding of PE’s effects.

Although acetylcholinesterase activity was assessed in pilot study, PE treatment did not significantly alter AChE activity. This finding suggests that PE does not exert its effect through inhibition of acetylcholine degradation. Instead, the observation that PE-induced facilitation of LTP was abolished by scopolamine implies that PE may act more directly on muscarinic acetylcholine receptor signaling pathways.

Although the present study focused on scopolamine-induced cholinergic dysfunction as a model of cognitive impairment, it is important to note that the cholinergic hypothesis alone cannot fully account for the complexity of Alzheimer’s disease, which also involves amyloid deposition, tau pathology, oxidative stress, and neuroinflammation. Therefore, our findings should be interpreted within this limitation, and future studies should examine the effects of PE on multiple pathological pathways beyond cholinergic neurotransmission.

In conclusion, our study demonstrates that PE improve scopolamine-induced cognitive impairment by enhancing synaptic plasticity through mAChR signaling. This effect is mediated by increased activity of CP-AMPARs; upregulation of synaptic proteins such as PSD-95, CaMKII, and NMDAR; and activation of downstream signaling pathways involved in LTP and memory formation. Although PE did not alter acetylcholine levels directly, it effectively restored cholinergic function impaired by scopolamine, suggesting a modulatory role on mAChR-mediated signaling. Furthermore, the cognitive enhancement observed across multiple behavioral tests, without affecting locomotor activity, supports the specificity of PE’s action on memory. Given the presence of rosmarinic acid and its known neuroprotective properties, PE may offer a promising therapeutic strategy for cognitive impairment associated with cholinergic dysfunction and synaptic deficits.

## 4. Materials and Methods

### 4.1. Materials

Scopolamine and IEM-1460 were purchased from Sigma-Aldrich (St. Louis, MO, USA). PE (Lot No. 240116-001) was provided by Borambio Co., Ltd. (Cheonan, Republic of Korea). *Perilla frutescens* var. *acuta* (L.) Britt. leaves were cultivated from Onnuri Agricultural Co. (Anseong, Republic of Korea). All reagents used for artificial cerebrospinal fluid (aCSF) preparation for electrophysiology were purchased from Sigma-Aldrich (St. Louis, MO, USA).

### 4.2. Preparation of PE

Ethanol extraction of *Perilla frutescens* var. *acuta* (L.) Britt. leaves was performed by S&D Co., Ltd. (Cheonan, Republic of Korea). Approximately 30 kg of *Perilla frutescens* var. *acuta* (L.) Britt. leaves were washed in a tank and subsequently dried. The washed *Perilla frutescens* var. *acuta* (L.) Britt. leaves were extracted by adding 60% alcohol solvent, and the extract was filtered using a cartridge filter and then concentrated under reduced pressure. After purifying the concentrate, an excipient (maltodextrin) was added to the purified solution, followed by spray-drying to prepare the PE (Lot No. 240116-001). PE was manufactured by extracting it twice with 60% ethanol at 40 °C for 2 h each, followed by concentration and freeze-drying. PE contained 7.5 ± 0.7 mg/g of rosmarinic acid (RA, Appendix A).

### 4.3. Animals

Experimental animals (male ICR mice, 6 weeks old) were purchased from Samtako (Osan, Republic of Korea). The ICR strain is commonly employed in febrile seizure and pharmacological studies due to its well-documented responsiveness and reproducibility. All animals were confirmed to be healthy, without evidence of immune dysfunction or prior experimental procedures. The mice were acclimated in the animal facility for one week before the experiments began. The animal facility maintained a 12 h light-dark cycle (7:00 am/ 7:00 pm), and temperature (23 ± 1 °C) and humidity (60 ± 5%) were kept constant. During both the acclimation and experimental periods, the animals had ad libitum access to food and water. Different animals were used for each behavioral experiment. For the passive avoidance test, 36 animals were used (6 per group). The Y-maze test included 54 animals (9 per group), while the Morris water maze test was conducted with 42 animals (7 per group). The animal sample size was determined using the G*Power software (version 3.1).

All animal experiments were conducted in accordance with NIH guidelines (NIH Publications No. 8023, revised 1978) and were approved by the Animal Ethics Committee of Daegu Hanny University (DHU2025-016). All experimental procedures were performed in accordance with the ARRIVE guidelines (https://arriveguidelines.org, accessed on 14 January 2025). Animals were randomly allocated to control and treatment groups using a random number table. Each animal was assigned a unique identification number, and the sequence of group allocation was determined according to the order of random numbers. This procedure ensured unbiased distribution across the groups. We did not control for potential confounders such as treatment order, measurement sequence, or cage location. However, to minimize experimental bias, a double-blind procedure was implemented, in which the individual administering the treatment and the experimenter conducting the behavioral assessments were different.

To minimize pain, suffering, and distress, all handling was performed gently, and the animals were monitored daily for changes in body weight, grooming, posture, or locomotor activity. No analgesics or anesthetics were required as no invasive procedures were performed. No unexpected adverse events occurred during the course of the study. Humane endpoints were established to reduce potential distress: animals showing signs of severe immobility, persistent abnormal posture, or inability to access food or water would have been removed from the study, but no animals met these criteria.

### 4.4. Acute Hippocampal Slices Preparation

The preparation of hippocampal slices and fEPSP recordings were performed as previously described [10]. We prepared acute mouse hippocampal slices for electrophysiology. Mice were anesthetized with isoflurane (3%), and their brains were rapidly removed and immersed in cold aCSF (NaCl, 124 mM; KCl, 3 mM; NaHCO_3_, 26 mM; NaH_2_PO_4_, 1.25 mM; CaCl_2_, 2 mM; MgSO_4_, 1 mM; D-glucose, 10 mM) for 1 min. Subsequently, the hippocampus was quickly isolated, and 400 µm thick hippocampal slices were prepared using a McIlwain tissue chopper. These slices were allowed to recover in aCSF at 28 °C with a mix gas for 2 h.

### 4.5. Electrophysiology

After recovery, the acute hippocampal slices were transferred to a recording chamber. The stimulating electrode was placed just below the CA2 region, and the recording electrode was positioned just below the CA1 region. Stimulation was applied every 30 sec to record field excitatory postsynaptic potentials (fEPSPs). Stimulation was adjusted to produce fEPSPs at approximately 50% of their maximum amplitude. Once stable fEPSP levels were obtained over a 20 min period, this was set as the baseline, and the stimulation intensity was varied to measure the slope of the fEPSP and the size of the presynaptic volley (input-output curve). Additionally, to measure the paired-pulse ratio, the interval between two stimuli was adjusted, and the ratio of these stimuli was recorded. Subsequently, another 20 min baseline was established, and high-frequency stimulation (HFS) was applied twice with a 30-sec interval to induce LTP. The size of LTP was compared using the average fEPSP during the last 5 min of recording. For blocking test, the hippocampal slices were cultured in aCSF containing PE for 2 h before recording. Scopolamine was administered in conjunction with PE, and IEM-1460 was applied during the baseline measurement. The measurement of fEPSPs was performed using the WinLTP (Ver. 2.20) program (https://www.winltp.com).

### 4.6. Passive Avoidance Test

The passive avoidance test was conducted following protocols established in our previous study [10]. For animal behavioral experiments, experimental animals that had been acclimated to the animal facility environment for one week were orally administered PE. Thirty min after PE (100, 250, and 500 mg/kg) administration, scopolamine (1 mg/kg) was administered intraperitoneally, followed by a training trial for the passive avoidance test 30 min later. The passive avoidance test was conducted using a two-chamber box. One chamber was white, and the other was black, with a guillotine door installed between them, allowing access to both chambers. The floors of both chambers were made of stainless steel grids, and the grid in the black chamber could deliver an electric shock. The white chamber had a light bulb. During the acquisition trial, the experimental animal was placed in the white chamber and allowed to acclimate for 10 s. Then, simultaneously with turning on the light, the guillotine door was opened. The time it took for the experimental animal to completely enter the black chamber was measured. If the animal did not move within 60 s, it was excluded from the experiment. Twenty-four hours later, a retention trial was conducted. The experimental animal was placed back in the white chamber, and 10 s later, the light was turned on, and the guillotine door was opened. The time it took for the experimental animal to completely enter the black chamber was measured.

### 4.7. Y-Maze Test

The Y-maze test was conducted following protocols established in our previous study [37]. Experimental animals, acclimated to the animal facility environment for one week, were orally administered PE (100, 250, and 500 mg/kg). Thirty minutes after PE administration, scopolamine (1 mg/kg) was administered intraperitoneally. Thirty minutes later, the Y-maze test was conducted. The Y-maze test was carried out in a maze with three arms diverging at 120-degree angles (45 × 5 × 3 cm). Each arm was designated as A, B, or C. The experimental animal was placed at the end of arm A, and the sequence of entries into each arm was recorded over a 8 min period. In this context, if the experimental animal entered all three different arms, it received a score of 1 point. The total number of entries and the score over the 8 min period were measured to calculate spontaneous alternation using the following formula: spontaneous alternation = Score/(Total arm entry − 2). In the Y-maze test, animals that exhibited fewer than 10 total arm entries were excluded from the experiment. two animals with total arm entries ≤ 10 were excluded; no animals were excluded from the other experiments.

### 4.8. Morris Water Maze Test

The Morris water maze test was conducted following protocols established in our previous study [37]. Experimental animals, acclimated to the animal facility environment for one week, were orally administered PE (100, 250, and 500 mg/kg). Thirty minutes after PE administration, scopolamine (1 mg/kg) was administered intraperitoneally. Thirty minutes later, the training session of the Morris water maze was conducted. The Morris water maze was performed using a cylindrical pool with a diameter of 45 cm. The pool was filled with water, artificially divided into four quadrants, and a platform was positioned in the center of one quadrant, located 0.5 cm below the water surface. To make the platform invisible, black food coloring was added to the water. During the training session, four trials were conducted each day for four days, with each trial lasting 1 min, to train the experimental animals to find the platform. The time it took for the experimental animals to find the platform from the starting point was measured. Each trial began with the experimental animal placed in a different quadrant of the pool. On the fifth day, the platform was removed, and the experimental animal was placed in the center of the pool to swim freely for 1 min. The time spent swimming in the quadrant where the platform had been located was measured (Ethovision Ver. 3.1.16, https://noldus.com/ethovision-xt, accessed on 19 February 2025). Animals that failed to swim or showed abnormal immobility during the training sessions were excluded.

### 4.9. Western Blot

Hippocampal tissues were homogenized in RIPA buffer containing protease and phosphatase inhibitors, then centrifuged at 12,500 rpm for 20 min. The supernatant was collected to extract total protein. Protein concentration was determined using a BCA assay kit (Thermo Fisher Scientific, Waltham, MA, USA). Protein samples were mixed with sample buffer, subjected to SDS-PAGE, and transferred to PVDF membranes. The membranes were blocked with 5% skim milk in TBST (0.1% Tween-20 in TBS) for 1 h, washed three times with TBST, and incubated overnight at 4 °C with primary antibodies. After washing, membranes were incubated with appropriate HRP-conjugated secondary antibodies at room temperature. Protein bands were visualized using an LAS Mini imaging system and quantified using ImageJ software (1.53k).

### 4.10. Statistics

The normality of the data was first assessed using the Shapiro–Wilk test. When the assumption of normality was met, the data were analyzed by one-way ANOVA, and Tukey’s post hoc test was applied for multiple comparisons. Post hoc comparisons were performed using the Tukey–Kramer test to account for unequal sample sizes among groups (Y-maze test). If normality was not satisfied, the Kruskal–Wallis test was employed as a non-parametric alternative. For comparisons between two groups, a Student t-test was conducted. For determining group interaction, two-way ANOVA was used followed by post hoc test using Bonferroni test was conducted. Data analysis was performed by researchers who were not directly involved in the experiments (Graphpad Prizm 6, https://www.graphpad.com/features, accessed on 17 November 2024). Data were presented as mean ± SEM, and statistical significance was indicated when *p* < 0.05.

## Figures and Tables

**Figure 1 ijms-26-09925-f001:**
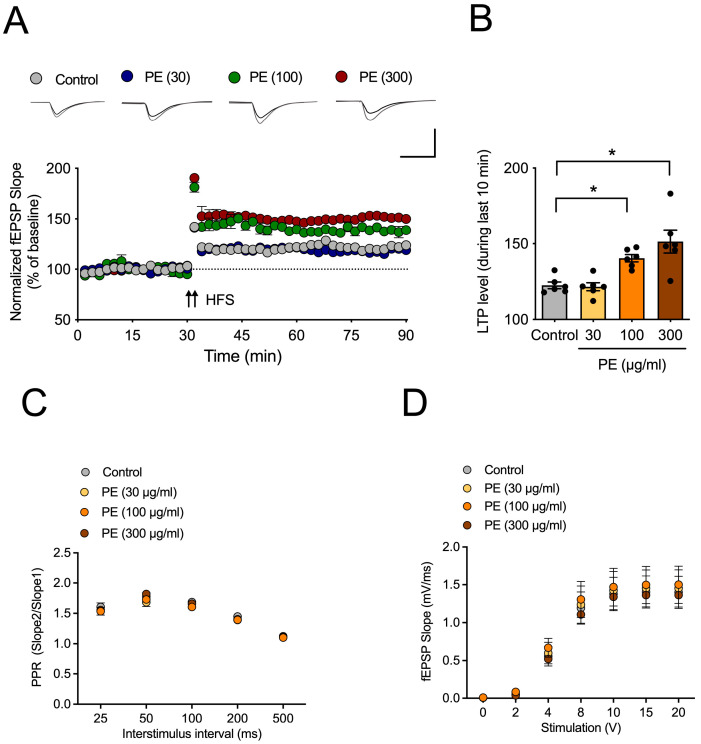
The effect of PE on hippocampal LTP. Acute hippocampal slices were incubated with PE (30, 100 or 300 μg/mL) for 2 h, and then fEPSP recordings were started. (**A**) The time-dependent changes in fEPSP in response to the same stimulus. LTP was induced by applying HFS (2 × 100 pulses, 100 Hz) after a stable response for 20 min. (**B**) The average value of fEPSP (field excitatory postsynaptic potential) during the last 10 min (yellow box in (**A**)). (**C**) PPR. (**D**) Input-output curve. Data represented as mean ± SEM. One way-ANOVA was used for statistics. * *p* < 0.05. HFS, high frequency stimulation. LTP, long-term potentiation. PE, standardized ethanol extract of *Perilla frutescens* var. *acuta* (L.) Britt. Leaves.

**Figure 2 ijms-26-09925-f002:**
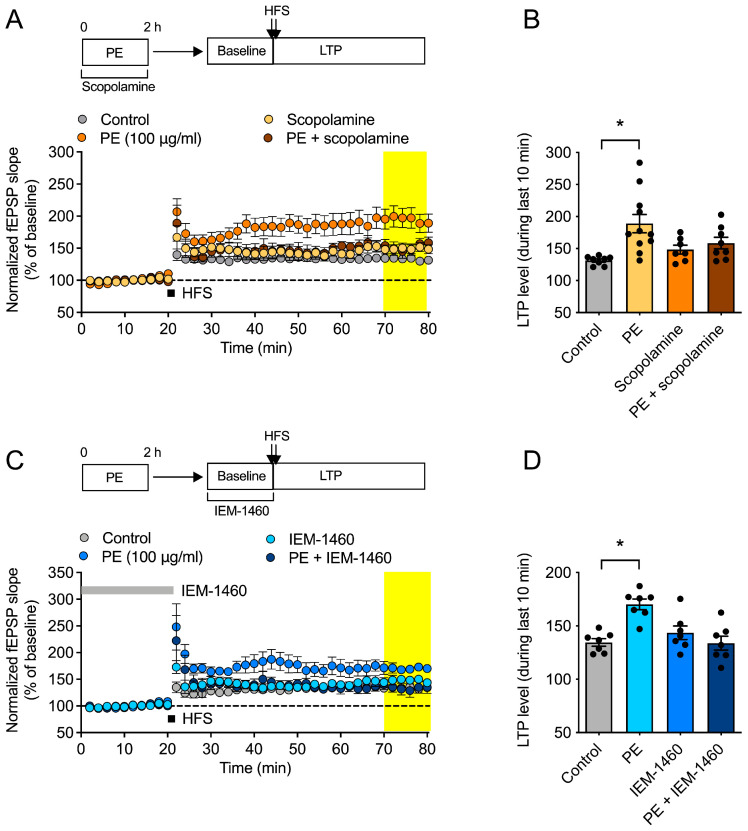
Possible mechanism of the effect of PE on LTP. (**A**,**B**) Effect of scopolamine on PE-facilitation of LTP. Acute hippocampal slices were incubated with PE (100 μg/mL) and scopolamine (1 μM) for 2 h, and then fEPSP recordings were started. (**A**) The time-dependent changes in fEPSP in response to the same stimulus. LTP was induced by applying HFS (2 × 100 pulses, 100 Hz) after a stable response for 20 min. (**B**) The average value of fEPSP (field excitatory postsynaptic potential) during the last 10 min (yellow box in (**A**)). (**C**,**D**) Effect of IEM-1460 on PE-facilitation of LTP. Acute hippocampal slices were incubated with PE for 2 h, and then fEPSP recordings were started. IEM-1460 (50 μM) was treated during baseline. (**C**) The time-dependent changes in fEPSP in response to the same stimulus. LTP was induced by applying HFS after a stable response for 20 min. (**D**) The average value of fEPSP (field excitatory postsynaptic potential) during the last 10 min (yellow box in (**C**)). Data represented as mean ± SEM. Two way-ANOVA was used for statistics. * *p* < 0.05. HFS, high frequency stimulation. LTP, long-term potentiation. PE, standardized ethanol extract of *Perilla frutescens* var. *acuta* (L.) Britt. Leaves.

**Figure 3 ijms-26-09925-f003:**
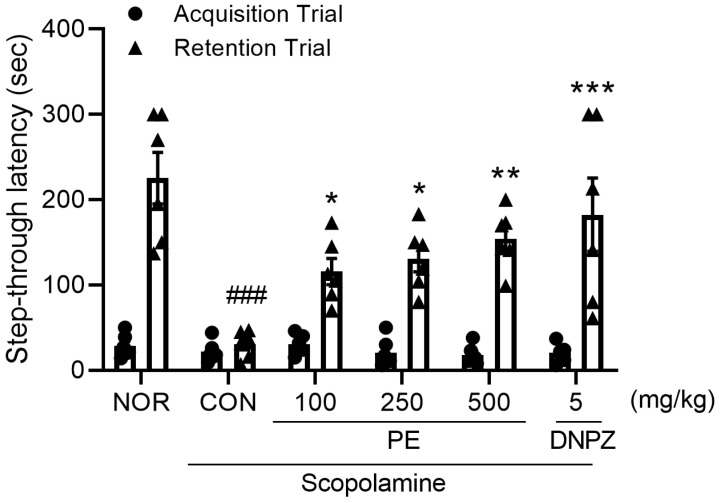
Ameliorating effect of PE on scopolamine-induced cognitive impairment in the passive avoidance test. PE was administered, followed by the administration of scopolamine 30 min later, and then behavioral tests were conducted 30 min later. Data represented as mean ± SEM. One way-ANOVA was used for statistics. ### *p* < 0.001 vs. normal group. * *p* < 0.05 vs. scopolamine group. ** *p* < 0.01 vs. scopolamine group. *** *p* < 0.001 vs. scopolamine group. NOR, normal group. CON, vehicle + scopolamine group. DNPZ, donepezil. PE, standardized ethanol extract of *Perilla frutescens* var. *acuta* (L.) Britt. Leaves.

**Figure 4 ijms-26-09925-f004:**
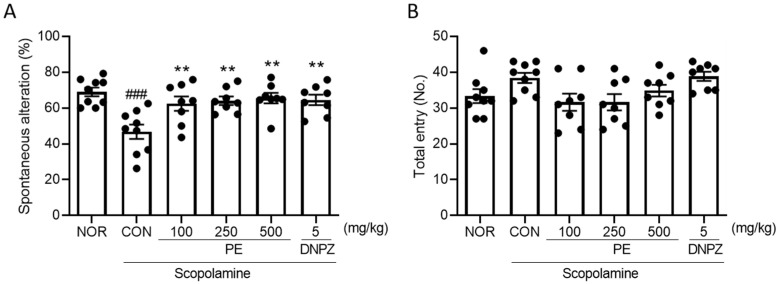
Ameliorating effect of PE on scopolamine-induced cognitive impairment in the Y-maze test. PE was administered, followed by the administration of scopolamine 30 min later, and then behavioral tests were conducted 30 min later. (**A**) spontaneous alternation. (**B**) Total arm entry. Data represented as mean ± SEM. One way-ANOVA was used for statistics. ### *p* < 0.001 vs. normal group. ** *p* < 0.01 vs. scopolamine group. NOR, normal group. CON, vehicle + scopolamine group. DNPZ, donepezil. PE, standardized ethanol extract of *Perilla frutescens* var. *acuta* (L.) Britt. Leaves.

**Figure 5 ijms-26-09925-f005:**
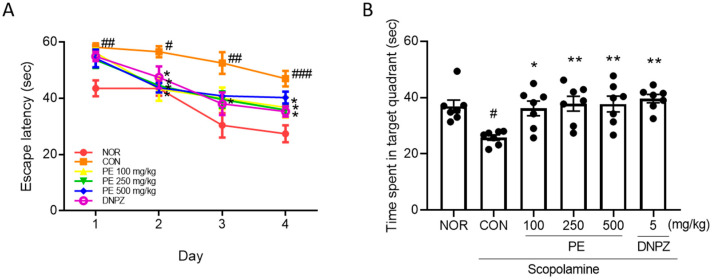
Ameliorating effect of PE on scopolamine-induced cognitive impairment in the Morris water maze test. During the experimental period, PE was administered daily, followed by the administration of scopolamine 30 min later, and then behavioral tests were conducted 30 min later. (**A**) Escape latency in training trial. (**B**) Escape latency in probe trial. Data represented as mean ± SEM. One way-ANOVA was used for statistics. # *p* < 0.05, ## *p* < 0.01, ### *p* < 0.001 vs. normal group. * *p* < 0.005, ** *p* < 0.01, *** *p* < 0.001 vs. scopolamine group. NOR, normal group. CON, vehicle + scopolamine group. DNPZ, donepezil. PE, standardized ethanol extract of *Perilla frutescens* var. *acuta* (L.) Britt. Leaves.

**Figure 6 ijms-26-09925-f006:**
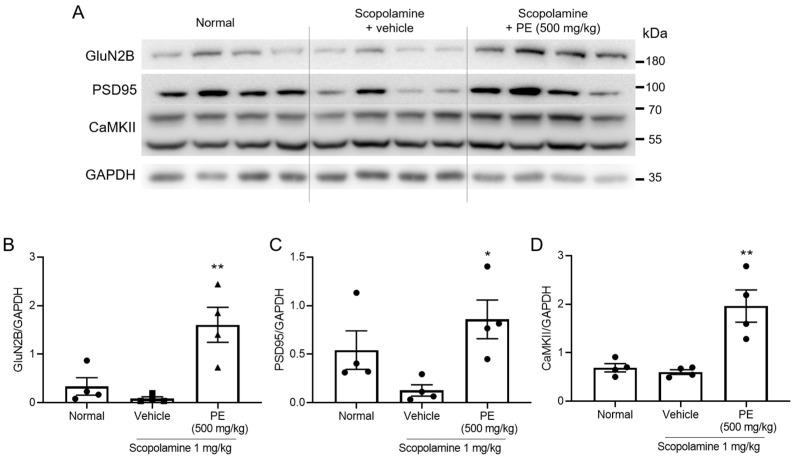
Effect of PE on synapse-related protein expression. During the experimental period, PE was administered daily, followed by the administration of scopolamine 30 min later, and then behavioral tests were conducted 30 min later. Immediately after the probe trial of Morris water maze, mouse hippocampus was isolated. (**A**) Immunoblots images of GluN2B, PSD-95, CaMKII and GAPDH. (**B**) Quantitative analysis of GluN2B/GAPDH. (**C**) Quantitative analysis of PSD-95/GAPDH. (**D**) Quantitative analysis of CaMKII/GAPDH. Data represented as mean ± SEM. One way-ANOVA was used for statistics. * *p* < 0.05 vs. vehicle group. ** *p* < 0.01 vs. vehicle group. PE, standardized ethanol extract of *Perilla frutescens* var. *acuta* (L.) Britt. Leaves.

**Figure 7 ijms-26-09925-f007:**
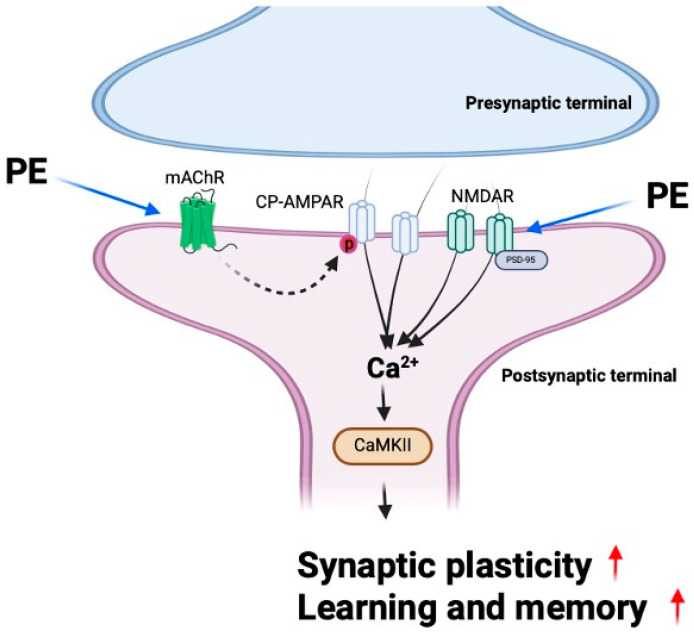
Graphical presentation of the effect of PE on synaptic function and learning and memory. PE stimulates mAChRs, enhancing the activity of CP-AMPARs, which leads to increased calcium influx, resulting in enhanced synaptic plasticity and improved memory. Additionally, PE upregulates excitatory synaptic markers such as NMDAR, PSD-95, and CaMKII, thereby promoting learning and memory enhancement.

## Data Availability

The raw data supporting the conclusions of this article will be made available by the authors on request.

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
