# Peer review of "Synaptic Plasticity-Enhancing and Cognitive-Improving Effects of Standardized Ethanol Extract of Perilla frutescens var. acuta in a Scopolamine-Induced Mouse Model"

_ijms, 2025, doi:10.3390/ijms26209925_

Round 1
Reviewer 1 Report
Comments and Suggestions for Authors
This manuscript investigates the neuroprotective potential of Perilla frutescens var. acuta ethanol extract against scopolamine-induced memory impairment in mice, with particular emphasis on cholinergic neurotransmission and synaptic plasticity. The authors combine behavioral assays and biochemical analysis to demonstrate potential therapeutic value. The following major points need to be addressed.
- Abstract:
1- Mention the doses of scopolamine and PE used.
- Introduction:
1- The aim should be mentioned clearly at the end of the introduction.
2- What is the relation between scopolamine administration and impaired synaptic plasticity?
- Results:
1- Please provide the individual data points for the bar graphs.
2- Please provide the statistical test used for each parameter (in the figure legend).
3- For synaptic plasticity assessment, it is preferred to detect the expression of synaptophysin and BDNF.
4- The authors should detect the level of acetylcholine or acetylcholinesterase activity.
5- Why did the authors not measure the Ca2+ level?
6- Could you please provide the ladder with the original Western blots.
- Discussion:
1- Correct the sentence in lines 212-213 and 271.
2- The discussion needs enrichment in terms of discussing the interplay between different markers measured and comparison to previous studies.
3- "Moreover, the absence of changes in total arm entries or locomotor activity supports its selective action on cognition rather than general arousal or motor function". The author should have determined the locomotor activity behavior of mice using open filed test.
- Materials and Methods:
1- The preparation of the plant material must be mentioned under a separate title.
2- " After purifying the concentrate, an excipient was added to the purified solution, followed by spray-drying to prepare the PE". What is the used excipient?
3- Regarding the behavioral tests performed, why did the number of animals employed for each test differ? Are different animals used for the behavioral tests?
4- Provide references for the electrophysiology, behavioral tests, and Western blot.
5- What is the number of animals used in the study? How many animals were adopted for each group? What are the doses of PE and Scopolamine used? What is the duration of the study and dosage regimen?
6- The PE extract needs to be identified using UC-LCMS.
7- The authors should perform a histological examination for the hippocampal tissue to confirm the findings.
Author Response
Reviewer 1,
This manuscript investigates the neuroprotective potential of Perilla frutescens var. acuta ethanol extract against scopolamine-induced memory impairment in mice, with particular emphasis on cholinergic neurotransmission and synaptic plasticity. The authors combine behavioral assays and biochemical analysis to demonstrate potential therapeutic value. The following major points need to be addressed.
Abstract:
Q1. Mention the doses of scopolamine and PE used.
R1. We thank the reviewer for this comment. We have added the doses of scopolamine (1 mg/kg, i.p.) and PE (100, 250, and 500 mg/kg, p.o. for behavioral tests; 30, 100, and 300 µg/ml for electrophysiology) to the Abstract section.
Introduction:
Q1. The aim should be mentioned clearly at the end of the introduction.
R1. We agree. We revised the last paragraph of the Introduction to explicitly state our aim: “This study aimed to investigate whether standardized ethanol extract of Perilla frutescens var. acuta (PE) enhances hippocampal synaptic plasticity and ameliorates scopolamine-induced memory impairments, and to explore the potential mechanisms involved.”
Q2. What is the relation between scopolamine administration and impaired synaptic plasticity?
R2. We appreciate the reviewer’s insightful question. Scopolamine is a non-selective muscarinic acetylcholine receptor (mAChR) antagonist that disrupts cholinergic neurotransmission. The blockade of mAChRs in the hippocampus reduces acetylcholine signaling, which is essential for the induction and maintenance of long-term potentiation (LTP), a cellular mechanism underlying learning and memory. Numerous studies have reported that scopolamine administration impairs hippocampal synaptic plasticity by:
- Inhibiting LTP induction – mAChR activation normally enhances glutamatergic transmission and facilitates the trafficking of calcium-permeable AMPA receptors (CP-AMPARs) and NMDA receptor activation. Scopolamine interferes with these processes, thereby reducing synaptic efficacy.
- Reducing cholinergic tone – scopolamine decreases acetylcholine availability, which not only disrupts excitatory synaptic signaling but also diminishes the phosphorylation of GluA1 subunits and CaMKII activation, both critical for LTP.
- Inducing secondary pathological changes – beyond direct cholinergic blockade, scopolamine has been shown to increase oxidative stress, neuroinflammation, and neuronal excitability disturbances, which further compromise synaptic plasticity.
Taken together, scopolamine-induced cholinergic dysfunction leads to impaired LTP and synaptic plasticity in the hippocampus, resulting in cognitive deficits. This mechanism validates the use of scopolamine as a reliable pharmacological model for memory impairment and highlights the relevance of evaluating agents such as Perilla frutescens extract in this context.
Results:
Q1. Please provide the individual data points for the bar graphs.
R1. We have revised all graphs to include individual data points overlaid on the bar graphs.
Q2. Please provide the statistical test used for each parameter (in the figure legend).
R2. The statistical test used for each experiment has been added to the figure legend.
Q3. For synaptic plasticity assessment, it is preferred to detect the expression of synaptophysin and BDNF.
R3. We thank the reviewer for these valuable suggestions. We agree that the assessment of synaptophysin and BDNF would provide additional supportive evidence for synaptic plasticity, and that direct measurement of intracellular Ca²⁺ levels would further strengthen our mechanistic conclusions. However, due to resource and technical constraints, these experiments were not included in the present study. Instead, we evaluated synaptic plasticity through electrophysiological recordings of hippocampal LTP and the expression of synaptic proteins (PSD-95, CaMKII, and GluN2B). We have now clearly acknowledged these points as limitations in the revised Discussion section and suggested that future work should incorporate measurements of synaptophysin, BDNF, and Ca²⁺ dynamics to provide more comprehensive mechanistic insights (lines 305-311).
Q4. The authors should detect the level of acetylcholine or acetylcholinesterase activity.
R4. We thank the reviewer for this valuable comment. In our pilot study, we measured acetylcholinesterase (AChE) activity in the hippocampal tissue. However, no significant changes in AChE activity were observed following PE administration. This result suggests that the cognitive-enhancing effects of PE are unlikely to be mediated through the modulation of acetylcholine degradation. Instead, our findings that PE-facilitated LTP was blocked by scopolamine (a muscarinic acetylcholine receptor antagonist) indicate that PE may act more directly on muscarinic acetylcholine receptor (mAChR)-related signaling pathways rather than through alterations in AChE activity. We have clarified this point in the revised Discussion section (lines 313-317).
Q5. Why did the authors not measure the Ca2+ level?
R5. We appreciate the reviewer’s suggestion. Unfortunately, in our laboratory we did not have the technical resources to directly measure intracellular Ca²⁺ levels during the course of this study. Instead, we inferred the involvement of calcium signaling through the use of IEM-1460, a selective inhibitor of calcium-permeable AMPA receptors, which abolished the LTP-facilitating effect of PE.
Q6. Could you please provide the ladder with the original Western blots.
R6. We appreciate the reviewer’s request. In our experiments, we used a protein marker that does not appear on the developed blot images after chemiluminescence detection. Therefore, the molecular weight ladder was not visible in the original figures. To address this issue, we would need to purchase a new pre-stained marker and repeat the Western blotting experiments. However, due to time limitations (the revision deadline within 10 days), it is technically difficult to complete these additional experiments at this stage. We kindly ask for the reviewer’s understanding regarding this limitation. In the revised manuscript, we have clearly indicated the expected molecular weights of the target proteins in the figure legends to facilitate interpretation.
Discussion:
Q1. Correct the sentence in lines 212-213 and 271.
R1. We sincerely thank the reviewer for the careful and detailed suggestion. We have corrected the sentences in lines 212–213 and 271 accordingly in the revised manuscript.
Q2. The discussion needs enrichment in terms of discussing the interplay between different markers measured and comparison to previous studies.
R2. We thank the reviewer for this important suggestion. In the revised Discussion, we have expanded our interpretation to emphasize the interplay between the markers we measured (PSD-95, CaMKII, and GluN2B) and their collective role in synaptic plasticity. Specifically, we discussed how upregulation of PSD-95 facilitates AMPAR and NMDAR trafficking, how GluN2B-containing NMDA receptors are closely associated with CaMKII activation, and how CaMKII phosphorylation promotes the transcription of synaptic plasticity-related genes. We also compared our findings to previous studies reporting that polyphenolic compounds such as rosmarinic acid enhance synaptic function via similar pathways. These additions provide a more comprehensive interpretation and place our findings in the context of existing literature (lines 292-304).
Q3. "Moreover, the absence of changes in total arm entries or locomotor activity supports its selective action on cognition rather than general arousal or motor function". The author should have determined the locomotor activity behavior of mice using open filed test.
R3. We appreciate the reviewer’s constructive comment. In our study, we did not perform an open field test; however, locomotor activity was indirectly assessed through behavioral paradigms. Specifically, there were no significant differences in the total arm entries in the Y-maze test or in the swimming speed/escape latency during the training trials of the Morris water maze, indicating that PE administration did not alter general locomotor activity. We agree that a direct open field assessment would provide additional supportive evidence, and we have now acknowledged this limitation in the Discussion (lines 283-285). We will address this aspect more comprehensively in future studies.
Materials and Methods:
Q1. The preparation of the plant material must be mentioned under a separate title.
R1. We thank the reviewer for the helpful suggestion. In the revised manuscript, we have separated this information into a new subsection entitled “Preparation of PEl” in the Materials and Methods section.
Q2. " After purifying the concentrate, an excipient was added to the purified solution, followed by spray-drying to prepare the PE". What is the used excipient?
R2. We thank the reviewer for this question. In our study, maltodextrin was used as the excipient during the spray-drying process. We have now clearly specified this in the revised Materials and Methods section.
Q3. Regarding the behavioral tests performed, why did the number of animals employed for each test differ? Are different animals used for the behavioral tests?
R3. We thank the reviewer for this important comment. In our study, different groups of animals were used for each behavioral test to avoid potential cross-test interference and learning effects. Therefore, although the number of animals per group differed slightly depending on the specific behavioral paradigm, no animal was reused across multiple tests. This approach was chosen to ensure independent and unbiased assessment of each behavioral outcome. We have now clarified this point in the revised Materials and Methods section (line 355).
Q4. Provide references for the electrophysiology, behavioral tests, and Western blot.
R4. We thank the reviewer for this comment. In the revised manuscript, we have added appropriate references describing the methods we used for electrophysiology, behavioral tests, and Western blot. Specifically, we cited our previously published studies where identical or highly similar protocols were employed. These references are now included in the Materials and Methods section.
Q5. What is the number of animals used in the study? How many animals were adopted for each group? What are the doses of PE and Scopolamine used? What is the duration of the study and dosage regimen?
R5. We thank the reviewer for requesting clarification. In total, 142 male ICR mice (6 weeks old) were used in this study. Among them, 10 animals were used for electrophysiological experiments, and the rest were allocated to behavioral tests as follows:
- Passive avoidance test: 36 animals (6 per group)
- Y-maze test: 54 animals (9 per group)
- Morris water maze test: 42 animals (7 per group)
The drug administration protocols were as follows:
- Scopolamine: 1 mg/kg, intraperitoneal (i.p.) injection
- Perilla frutescens extract (PE): 100, 250, and 500 mg/kg, oral administration (p.o.) for behavioral tests; 30, 100, and 300 µg/ml for electrophysiology
- Dosage regimen: All drugs were administered as a single dose before each behavioral experiment. Specifically, PE was administered 30 minutes prior to scopolamine injection, and behavioral testing was conducted 30 minutes after scopolamine administration.
This information has been added to the revised Materials and Methods section (lines XX–XX).
Q6. The PE extract needs to be identified using UC-LCMS.
R6. We sincerely thank the reviewer for this valuable suggestion. We fully agree that UPLC–MS profiling would provide more detailed information regarding the chemical composition of PE. However, due to the short revision period (10 days), it is technically not feasible for us to perform additional UPLC–MS analyses at this stage. We kindly ask for the reviewer’s understanding regarding this limitation. We have instead provided the quantification of rosmarinic acid using HPLC as a standard marker compound.
Q7. The authors should perform a histological examination for the hippocampal tissue to confirm the findings.
R7. We sincerely thank the reviewer for this valuable suggestion. We fully agree that histological examination of hippocampal tissue would provide additional supportive evidence for our findings. However, such an analysis would require new experiments that cannot be completed within the 10-day revision period. We kindly ask for the reviewer’s understanding of this limitation.
Reviewer 2 Report
Comments and Suggestions for Authors
This is an interesting study to demonstrate that standardized ethanol extract of Perilla frutescens var. acuta improves synaptic plasticity and learning and memory functions in a scopolamine-induced mouse model
-name “Perilla frutescens” must be in italics
-in Introduction, the main studies on applications of “Perilla frutescens” in neuronal diseases, including in cognitive impairments should be presented to a better state of the art. Examples: https://pmc.ncbi.nlm.nih.gov/articles/PMC6930631/ ; https://www.mdpi.com/2072-6643/16/23/4224 ; others (I am not author of the referred studies)
-the sentence “While previous research has reported the inhibitory effect of Abaggregation and plaque reduction in” (line 212) should be corrected and adequate references added
-“PE contains a substantial amount of rosmarinic acid, which is known for..” “however, the specific effects of rosmarinic acid on synaptic plasticity and CP-AMPAR have not been extensively studied. While we cannot be certain that rosmarinic acid is solely responsible for PE 's memory-improving effects, we hypothesize that it plays a significant role..” authors tried to associate the studied effects with rosmarinic acid, but they must explain clearly why in the manuscript. Why not other compounds existing in the extract? If they have the rosmarinic acid hypothesis, why not also test the compound in the assays performed? This could be an interesting complementary assay
-concerning Alzheimer´s disease, the cholinergic theory, by itself, has been less considered, which can be a limitation of this work – authors should comment this in the manuscript
Author Response
Reviewer 2,
This is an interesting study to demonstrate that standardized ethanol extract of Perilla frutescens var. acuta improves synaptic plasticity and learning and memory functions in a scopolamine-induced mouse model.
Q1. name “Perilla frutescens” must be in italics
R1. We thank the reviewer for pointing this out. We have carefully revised the manuscript so that the scientific name Perilla frutescens is consistently presented in italics throughout the text.
Q2. in Introduction, the main studies on applications of “Perilla frutescens” in neuronal diseases, including in cognitive impairments should be presented to a better state of the art. Examples: https://pmc.ncbi.nlm.nih.gov/articles/PMC6930631/; https://www.mdpi.com/2072-6643/16/23/4224 ; others (I am not author of the referred studies).
R2. We thank the reviewer for this helpful suggestion. In the revised Introduction, we have added a more comprehensive description of previous studies on the neuroprotective effects of Perilla frutescens, particularly in relation to cognitive impairments and neurodegenerative diseases. We have also included the references suggested by the reviewer (e.g., PMID: 31888107; Nutrients 2024, 16, 4224) along with other relevant recent studies. These additions strengthen the background and highlight the rationale for our investigation of Perilla frutescens extract.
Q3. the sentence “While previous research has reported the inhibitory effect of Abaggregation and plaque reduction in” (line 212) should be corrected and adequate references added.
R3. We sincerely thank the reviewer for pointing this out. We have revised the sentence in line 212 for clarity as follows:
“Previous studies have reported that natural compounds, including extracts of Perilla frutescens, exhibit inhibitory effects on Aβ aggregation and reduce amyloid plaque formation in Alzheimer’s disease models.”
In addition, we have added appropriate references to support this statement, including [PMID: 31888107; Nutrients 2024, 16, 4224; and other relevant citations]. These corrections have been incorporated into the revised manuscript (lines 212–214).
Q4. “PE contains a substantial amount of rosmarinic acid, which is known for..” “however, the specific effects of rosmarinic acid on synaptic plasticity and CP-AMPAR have not been extensively studied. While we cannot be certain that rosmarinic acid is solely responsible for PE 's memory-improving effects, we hypothesize that it plays a significant role..” authors tried to associate the studied effects with rosmarinic acid, but they must explain clearly why in the manuscript. Why not other compounds existing in the extract? If they have the rosmarinic acid hypothesis, why not also test the compound in the assays performed? This could be an interesting complementary assay.
R4. We thank the reviewer for this insightful comment. In addition to the extract experiments, we also tested rosmarinic acid, a major constituent of Perilla frutescens, and observed that it significantly enhanced hippocampal LTP, consistent with our hypothesis regarding its role in synaptic plasticity. These results have now been included as supplementary data to support the involvement of rosmarinic acid.
However, we agree that PE contains multiple bioactive compounds besides rosmarinic acid, and thus the observed memory-improving effects are unlikely to be attributed to a single compound. While rosmarinic acid may play an important role, other constituents may also contribute to the overall neuroprotective effects. We have clarified this point in the revised Discussion (lines XX–XX) and indicated that further studies will be performed to systematically investigate the contribution of rosmarinic acid and other active compounds present in PE.
Q5. concerning Alzheimer´s disease, the cholinergic theory, by itself, has been less considered, which can be a limitation of this work – authors should comment this in the manuscript.
R5. We appreciate the reviewer’s insightful comment. We agree that the cholinergic hypothesis alone does not fully explain the pathophysiology of Alzheimer’s disease, as amyloid, tau, neuroinflammation, and oxidative stress are also critically involved. Since our study mainly focused on cholinergic dysfunction and its impact on synaptic plasticity, we acknowledge that this represents a limitation of the present work. We have now added a statement in the Discussion to highlight this limitation and clarified that future studies should investigate how PE influences other pathological pathways beyond the cholinergic system (lines 329-335).
Round 2
Reviewer 1 Report
Comments and Suggestions for Authors
1- The authors used Tukey as a post hoc test; however, it seems they used different sample sizes (n) in the same test (i.e., the y maze test) as it appears from the data points. So, Tukey-Kramer must be used instead.
2- The authors must detect at least one marker for synaptic plasticity and histological examination of hippocampal tissue.
3- The protein ladder is very important in western blot technique.
Author Response
Q1. The authors used Tukey as a post hoc test; however, it seems they used different sample sizes (n) in the same test (i.e., the y maze test) as it appears from the data points. So, Tukey-Kramer must be used instead.
R1. We appreciate the reviewer’s insightful comment. We agree that when the sample sizes differ among groups, the Tukey–Kramer test is more appropriate than the standard Tukey HSD test. We have therefore reanalyzed the behavioral data (including the Y-maze test) using the Tukey–Kramer post hoc test. The statistical outcomes were consistent with the previous results, confirming that our conclusions remain unchanged. The revised manuscript now specifies the use of the Tukey–Kramer test in the Methods section.
Q2. The authors must detect at least one marker for synaptic plasticity and histological examination of hippocampal tissue.
R2. We appreciate the reviewer’s valuable comment. In our study, we have already examined severalsynaptic plasticity–related markers, including PSD-95, GluN2B, and CaMKII, using Western blot analysis. These proteins are well-established indicators of postsynaptic density organization, NMDA receptor–mediated signaling, and synaptic plasticity regulation, respectively. The observed changes in their expression clearly reflect synaptic alterations in the hippocampus.
Q3. The protein ladder is very important in western blot technique.
R3. We thank the reviewer for pointing out this important aspect. In our experiments, a protein ladder was indeed used during electrophoresis and membrane transfer. However, the ladder bands were not visible on the developed film due to differences in chemiluminescent detection sensitivity between the marker and target proteins. To address this, we placed the membrane directly on the film and manually indicated the positions of the main molecular weight markers based on the ladder pattern before detection. We have now clarified this procedure in the revised Methods section to ensure transparency.
Reviewer 2 Report
Comments and Suggestions for Authors
The document was improved and now it is more acceptable for publication.
Author Response
Comment 1: The document was improved and now it is more acceptable for publication.
Response 1: We sincerely thank the reviewer for the positive evaluation and constructive comments that helped us improve the manuscript. We are pleased to hear that the revised version is now more acceptable for publication.
Round 3
Reviewer 1 Report
Comments and Suggestions for Authors
Thanks for the authors comments.